# Effects of Ketogenic Diet on Quality of Life in Parkinson Disease: An Integrative Review

**DOI:** 10.3390/nu17213343

**Published:** 2025-10-24

**Authors:** Maria Giulia Golob, Stefano Mancin, Diego Lopane, Chiara Coldani, Daniela Cattani, Alessandra Dacomi, Giuseppina Tomaiuolo, Fabio Petrelli, Giovanni Cangelosi, Simone Cosmai, Alice Maria Santagostino, Beatrice Mazzoleni

**Affiliations:** 1Department of Biomedical Sciences, Humanitas University, Via Rita Levi Montalcini 4, Pieve Emanuele, 20090 Milan, Italy; maria.golob@st.humanitas.eu (M.G.G.); stefano.mancin@humanitas.it (S.M.); chiara.coldani@humanitas.it (C.C.); daniela.cattani@humanitas.it (D.C.); alessandra.dacomi@humanitas.it (A.D.); giuseppina.tomaiuolo@humanitas.it (G.T.); simone.cosmai@gavazzeni.it (S.C.);; 2IRCCS Humanitas Research Hospital, Via Manzoni 56, Rozzano, 20089 Milan, Italy; alice.santagostino@humanitas.it; 3Experimental Medicine and Public Health Unit, School of Pharmacy, University of Camerino, Via Madonna delle Carceri 9, 62032 Camerino, Italy; fabio.petrelli@unicam.it

**Keywords:** neurodegeneration, non-motor symptoms, nutritional intervention, ketosis, neuroprotection, integrative review

## Abstract

*Background/Aims*: Parkinson’s disease (PD) is a progressive neurodegenerative disorder caused by the degeneration of dopaminergic neurons, leading to motor and non-motor symptoms that significantly impair quality of life (QoL). Oxidative stress (OS) and neuroinflammation play a key role in its progression. The ketogenic diet (KD) may have neuroprotective effects by reducing these factors through ketosis. The primary aim of this narrative review is to examine the impact of the ketogenic diet on the quality of life and symptomatology of patients with PD, evaluating its effects on motor and non-motor symptoms, as well as on certain metabolic parameters. Secondary aims included assessing the feasibility of and adherence to the diet, as well as its tolerability and safety. *Methods*: A search of PubMed, Scopus, Embase, CINAHL and Cochrane databases up to June 2025 was performed. Eligible studies included adults with PD following a KD regimen. Data were extracted regarding QoL outcomes, adverse events, and risk of bias included for synthesis. *Results*: A total of 152 patients were included across 6 studies. KD showed a small to moderate effect size on QoL improvements, particularly in non-motor domains such as fatigue and sleep quality. However, findings were inconsistent across studies. Risk of bias was rated moderate to high due to small sample sizes, heterogeneous methodologies, and lack of blinding. The most frequently reported adverse events were gastrointestinal disturbances (nausea, constipation), weight loss, and transient fatigue. *Conclusions*: Although preliminary evidence suggests a potential benefit of KD on QoL in PD patients, the small number of participants, short follow-up, and high heterogeneity significantly limit generalizability. Further large, controlled trials with rigorous methodology are warranted before relevant conclusion benefits can be drawn.

## 1. Introduction

Parkinson’s disease (PD) is a progressive neurodegenerative disorder characterized by the loss of dopaminergic neurons in the substantia nigra, leading to motor and non-motor symptoms that markedly impair patients’ quality of life (QoL) [1,2]. In addition to motor manifestations such as bradykinesia, tremor, and rigidity, PD patients frequently experience non-motor features including cognitive decline, sleep disturbances, mood disorders, and malnutrition, all of which further worsen their overall well-being [3,4,5,6]. Globally, PD represents one of the most prevalent neurodegenerative conditions, with a rising incidence due to population aging, and it is expected to double by 2040 [7,8,9]. This growing burden has significant social and economic implications for healthcare systems [10,11,12]. Among the potential non-pharmacological approaches to support PD management, nutritional strategies have gained increasing attention. The ketogenic diet (KD), a high-fat and low-carbohydrate regimen, induces ketosis, which may reduce oxidative stress (OS) and inflammation through β-hydroxybutyrate–mediated mechanisms [13,14,15]. In addition to classical KD, other regimens such as intermittent fasting, caloric restriction, and time-restricted feeding can also induce ketogenesis and have been investigated for their possible neuroprotective effects [16]. In recent years, several systematic reviews and meta-analyses have examined KD and related dietary interventions in PD and other neurodegenerative diseases, mostly focusing on motor outcomes or biochemical markers rather than QoL [17,18,19,20]. However, the heterogeneity and methodological limitations of existing studies make it difficult to draw firm conclusions. The KD, characterized by high fat and low carbohydrate intake, induces ketosis that yields inflammation and oxidative stress reduction via β-hydroxybutyrate action [21,22]. Additionally, it can also activate antioxidant genes and reduce pro-inflammatory cytokines [23], potentially exerting neuroprotective effects that could benefit PD patients (Figure 1). From a clinical and assistive practice standpoint, nutritional strategies such as KD may constitute feasible adjunctive interventions within the therapeutic framework for PD, with potential to complement pharmacological treatments and to enhance patients’ overall QoL [17,18,19,20]. Nevertheless, despite growing scientific interest, the available evidence remains heterogeneous and methodologically limited, with insufficient emphasis on patient-centered outcomes such as QoL, thereby underscoring the need for more robust and clinically oriented investigations.

### 1.1. Aims and Research Questions

#### 1.1.1. Primary

Aim: To evaluate the effect of the KD on QoL and symptomatology in patients with PD, considering both motor and non-motor symptoms as well as selected metabolic parameters.

-Does the KD improve QoL and reduce motor and non-motor symptoms in patients with PD, while also producing measurable changes in specific metabolic parameters?

#### 1.1.2. Secondary

Aim: To assess the feasibility and adherence to a ketogenic regimen in patients with PD, together with its tolerability and safety profile.

-Is the KD feasible and sustainable for patients with PD in terms of adherence, and what is its tolerability and safety profile (including the most common adverse events)?

## 2. Methods

### 2.1. Study Design & Protocol Registration

This integrative review was conducted according to the methodological framework of Whittemore & Knafl [24]. For transparency, the protocol of this review has been registered in the Open Science framework database and is available at: https://doi.org/10.17605/OSF.IO/R9XFA.

### 2.2. Problem Identification

The research question was structured according to the PICO framework [25]. For the purpose of this review: P = PD patients; I = KD; O = QoL improvement, symptomatology reduction. As a secondary outcome we addressed the feasibility of the intervention on the identified population. No comparison intervention was specified.

### 2.3. Inclusion and Exclusion Criteria

In this review, primary studies have been considered. Inclusion criteria singled out as eligible those studies conducted exclusively on patients who only suffered from PD, at any stage. These evaluated the effect of the KD on patients’ QoL and symptomatology, also compared to other nutritional approaches. As a consequence, we excluded articles which target population was affected by another degenerative diseases and based on animal models, or which intervention solely consisted of a diet that could not be traced back to the ketogenic one. However, regimens attributable to the ketogenic model were included, such as low-carbohydrate/high-fat diets (LCHF) and exogenous ketone supplementation, in particular medium chain triglyceride (MCT). Based on the research question and review objective, specific scales, symptomatology and metabolic biomarkers were considered to verify improvement in QoL.

### 2.4. Literature Search

A comprehensive literature search was conducted up to June 2025. Major scientific databases, specifically PubMed-Medline, Embase, Cochrane Library, and CINAHL were searched. The search strategy included key terms such as “ketogenic diet,” “Parkinson’s disease,” and “quality of life,” along with relevant synonyms and related phrases. Mesh terms associated with keywords and free terms were also used both in PubMed and Cochrane Library databases to create search strings. Boolean operators (AND/OR) were carefully applied to relate Mesh terms, equivalent terms, and keywords, ensuring a broad yet targeted search. Once the search results were retrieved from each database, a selection process was carried out using Rayyan software (https://www.rayyan.ai/, Copyright © 2025 RAYYAN). The screening was conducted based on specific criteria. Initially, duplicates were identified and excluded. The remaining studies underwent an additional screening process to assess their alignment with the research question. The screening involved selection based on titles, followed by evaluation of the abstract and full-text content to determine relevance to the PICO framework and the inclusion/exclusion criteria established at the outset of this review (Appendix A).

### 2.5. Evaluation of Risk of Bias and Methodological Quality of Studies Included

To rigorously assess the methodological quality and relevance of the selected studies, we used the Joanna Briggs Institute (JBI) Critical Appraisal Tools. Depending on the type of study, specific tools were used: randomized controlled trial (RCT) studies [26], pilot studies [27], and longitudinal studies [28]. Scores were derived as the percentage of ‘Yes’ responses [26,27]. The quasi-experimental checklist was used for pre–post designs. Based on a previous study [29], studies with a JBI score ≥ 70% were classified as high quality, those with a score between 70% and 50% as medium quality, and those with a score < 50% as low quality (Appendix A).

### 2.6. Data Extraction and Synthesis

Data from the selected articles were extracted and reported in tables, capturing the following information: authors, study design, sample, objective, intervention and duration of intervention application, outcome, outcome measurement, limitations, assessment of quality/risk of bias. The records included in this integrative review were systematically classified according to intervention type and outcome and reported as a narrative summary supplemented by figures and tables.

## 3. Results

A total of six studies met the inclusion criteria [30,31,32,33,34,35], comprising four RCTs, one pilot feasibility study, and one longitudinal study (Figure 2). In total, 527 records were identified through database searches: PubMed (n = 180), Embase (n = 140), Cochrane Library (n = 92), and CINAHL (n = 115). After removing 132 duplicates, 395 titles and abstracts were screened. Of these, 368 were excluded because they involved non-PD populations, non-ketogenic or mixed nutritional interventions, or animal models. The remaining 27 full-text articles were assessed for eligibility; 21 were excluded mainly due to lack of QoL outcomes (n = 10), absence of a ketogenic component (n = 6), or overlapping data (n = 5). Six studies met all inclusion criteria and were included in the final synthesis. Collectively, they enrolled fewer than 150 patients, with follow-up durations ranging from 3 weeks to 6 months (Summary Table 1 and Table 2).

Although most studies applied acceptable methodological frameworks according to JBI checklists, the overall quality remains limited due to small sample sizes, short follow-up, and open-label or quasi-experimental designs, which inevitably increase the risk of bias. Considerable heterogeneity was also observed in dietary protocols (classical KD, MCT-supplemented KD, modified Atkins), comparator regimens, outcome measures, and patient characteristics. A critical synthesis of the main clinical and metabolic findings is presented in Table 2.

Across trials, KD consistently induced weight reduction and Body Mass Index (BMI) decline, with some also reporting waist circumference reduction [30,31,32,33,34,35]. Improvements in glycemic control [glycated hemoglobin (HbA1c) and glucose variability] emerged in selected studies [33,34,35]. Lipid responses were inconsistent: triglycerides decreased and High Dose Lipoprotein (HDL) increased in several reports [33,34,35], whereas one RCT also found significant elevations in Low Dose Lipoprotein (LDL) and total cholesterol [30]. These results suggest a physiological response to carbohydrate restriction, but the variability across studies prevents generalization. Motor outcomes, measured with United Parkinson’s Disease Rating Scale (UPDRS) or Movement Disorders Society-Unified Parkinson’s Disease Rating Scale (MDS-UPDRS), showed only marginal or inconsistent benefits. Most studies [30,31,33,34,35] did not demonstrate significant changes in bradykinesia, tremor, or rigidity, except for Koyuncu et al. [32], who observed improvements in voice-related motor function (VHI-10). The modest effects described may therefore reflect nonspecific factors such as weight loss or placebo responses rather than a true disease-modifying action. By contrast, more consistent signals were found for non-motor symptoms. Phillips et al. [30] and Tidman et al. [34,35] reported reductions in depression and anxiety scores with Parkinson Anxiety Scale (PAS) or Center for Epidemiologic Studies Depression Scale Revised (CESD-R-20), while Krikorian et al. [31] found improvements in lexical access and verbal memory. Choi et al. [33] described modest reductions in NMSS scores without effects on working memory. Since non-motor symptoms are major determinants of health-related quality of life (HRQoL) [36,37], these preliminary data suggest a potentially meaningful contribution of KD in this domain. Feasibility was generally high, with dropout rates between 0 and 14%. Adherence was supported by caregiver involvement, education, and practical tools such as cookbooks and shopping lists [33,34,35], and was confirmed in most cases by ketone monitoring [30,33,34]. Nonetheless, feasibility in routine practice remains uncertain, as participants were followed in controlled settings with intensive support. Adverse events were usually mild and transient, consisting mainly of gastrointestinal complaints, dizziness, headache, and fatigue [30,31,32,33]. These are consistent with the so-called “keto flu” [38] and generally resolved with supportive measures. No severe adverse events were reported, but the short observation periods preclude any conclusions about long-term safety, especially concerning renal and hepatic function [39]. Taken together, the evidence suggests that KD may exert modest benefits on metabolic control and non-motor symptoms, while effects on motor outcomes remain weak. However, heterogeneity, small sample sizes, and methodological limitations considerably restrict the robustness and generalizability of these findings.

## 4. Discussion

The KD has recently drawn attention as a potential therapeutic approach for PD, owing to its ability to modulate energy metabolism and reduce neuroinflammation through the production of ketone bodies [40]. While preclinical mechanisms suggest a neuroprotective role, current clinical evidence remains weak and heterogeneous. In particular, improvements in motor outcomes are not consistently supported. Although some studies reported slight benefits in bradykinesia, tremor, and rigidity [30,33,34,35], these effects were modest and not sufficient to establish a causal link. Weight loss [41] and placebo response induced by comprehensive educational protocols [30,34,35] may have influenced the findings, as previously reported for PD motor outcomes [42]. Some authors have hypothesized that longer adaptation to ketosis is necessary for clinically relevant effects [43], yet even a six-month intervention failed to demonstrate significant improvements [35]. Thus, evidence to date does not justify considering KD as an effective strategy for motor symptom control. More promising data concern non-motor manifestations, which are among the most disabling features of PD and exert the greatest impact on HRQoL [36]. None of the included studies performed statistical adjustments for potential confounding factors such as body-weight reduction, increased physical activity, or placebo effects derived from the intensive educational and monitoring support provided. Consequently, improvements in non-motor symptoms may in part reflect nonspecific effects rather than a direct physiological action of the ketogenic diet. Future RCTs should incorporate adequate control arms and adjust for these variables to isolate true dietary effects. Conventional pharmacological therapies are often ineffective against depression, anxiety, apathy, and cognitive impairment [37], and in this domain the KD appears to exert more consistent benefits. Several trials reported statistically significant improvements in anxiety and depressive symptoms [32,34,35], with anxiety responding within weeks [34] and depression after longer follow-up [35], in line with evidence from other non-pharmacological interventions [44,45]. Encouraging results also derive from studies of KD in psychiatric and mood disorders [46,47]. Cognitive improvements, particularly in lexical access and verbal memory, have been reported [31] and are consistent with findings in Alzheimer’s disease [48]. These results, although preliminary, suggest a potential role for KD as an adjunctive intervention targeting non-motor symptoms and overall well-being. Adherence and feasibility are critical aspects in dietary interventions. The reviewed studies generally reported high compliance [32,33,34,35], supported by structured education of patients and caregivers [33,35] and practical tools such as cookbooks and shopping lists [30,34,35]. MCT supplementation further enhanced adherence, reducing the need for rigid dietary restriction [33,49], as previously observed in epilepsy [50]. Monitoring through food diaries and blood ketone tracking [30,33,34] proved reliable and feasible, although the small, motivated populations involved may not reflect real-world adherence. In terms of safety, KD was overall well tolerated. Reported adverse events, including fatigue, dizziness, “brain fog,” and gastrointestinal disturbances, were mostly mild and transient [31,32,38]. These manifestations are generally attributable to the “keto flu” and resolve with hydration and fiber adjustment [38]. Importantly, the short follow-up of the reviewed studies does not allow conclusions on long-term safety, particularly concerning renal and hepatic function, which may be challenged by sustained ketone metabolism [51,52,53]. All studies consistently reported weight loss in KD participants [30,31,33,34,35], accompanied by reductions in BMI and waist circumference. Only Choi et al. (2024) [33] assessed body composition, reporting a modest decrease in lean mass after three weeks of KD despite stable body weight in some participants. None of the included trials implemented strategies to preserve muscle mass, such as resistance training or optimized protein intake. Considering the high prevalence of sarcopenia and frailty in PD, future dietary protocols should integrate exercise and nutritional support to mitigate potential muscle loss. In overweight patients these changes were associated with psychological benefits and modest mobility improvements [30,33,34,35]. Nevertheless, the loss of lean mass reported in one study [33] deserves attention, as sarcopenia may exacerbate disability in PD [52,53]. These findings highlight the importance of monitoring nutritional status, as patients with neurological disorders such as PD are at increased risk of malnutrition, which may further compromise functional capacity and QoL [54,55]. Lipid profiles showed reductions in triglycerides and increases in HDL [30,33,34,35], though one study also reported increases in LDL and total cholesterol [30], highlighting the importance of fat quality, with a preference for unsaturated over saturated fatty acids [56]. The variability observed across lipid outcomes likely depends on the quality and composition of dietary fats. Most studies used mixed fat sources without specifying the relative contribution of saturated versus unsaturated fatty acids. Trials emphasizing medium-chain triglycerides (MCT) or unsaturated fats reported neutral or favorable lipid profiles, whereas those richer in saturated fats showed elevations in LDL and total cholesterol. This finding underscores the importance of fat quality in designing safer ketogenic protocols for PD patients. Improvements in glycemic control were also observed [30,33,34,35], plausibly linked to carbohydrate restriction [57]. An underexplored issue is the interaction between KD and pharmacotherapy. Beyond the single pharmacokinetic study showing no major change in levodopa plasma levels [58,59], theoretical mechanisms may affect drug response. Ketogenic diets can alter gastric emptying and intestinal motility, modify gut microbiota composition, and increase competition between dietary amino acids and levodopa for intestinal transporters. These factors could potentially influence absorption and lead to fluctuations in motor performance. Clinicians should monitor for changes in “on–off” response patterns, gastrointestinal tolerance, and hydration status when initiating KD in patients under dopaminergic therapy. Further research is warranted to elucidate these interactions. Furthermore, KD-induced alterations in gastrointestinal motility and microbiota, observed in epilepsy [60], could also affect drug response. These aspects require further investigation, given the clinical relevance of motor fluctuations in PD. Overall, the current body of evidence must be interpreted with caution. The reviewed studies are characterized by small sample sizes, heterogeneous protocols, open-label or pre–post designs, and short treatment durations [30,31,32,33,34,35]. These methodological weaknesses introduce a high risk of bias and limit the generalizability of findings. While KD appears safe and feasible in the short term, its clinical role in PD remains experimental.

### 4.1. Future Perspectives

Future research should prioritize adequately powered randomized controlled trials with standardized protocols, longer follow-up, and validated QoL measures. Comparative studies of different ketogenic regimens, incorporation of flexible dietary models, and systematic ketonemia monitoring will be essential to optimize adherence. Safety assessment should include renal, hepatic, and metabolic and metabolomics monitoring, and special attention should be devoted to the impact on non-motor symptoms [61,62,63,64]. In addition, alternative approaches such as intermittent or prolonged fasting merit exploration as potentially more acceptable strategies [65,66,67,68,69]. Ultimately, integrating nutritional interventions within multidisciplinary care, together with the support of emerging technologies such as artificial intelligence or mobile applications health technology in general, already applied successfully in other chronic diseases, may provide new opportunities to improve management and QoL for PD patients [70,71,72,73,74,75,76,77,78,79,80].

### 4.2. Limitations

This study presents several limitations. First, the methodological design based on an integrative review entails a risk of heterogeneity and subjectivity in the selection and synthesis of evidence. Although a proven method was applied to ensure transparency and rigor, the absence of a systematic procedure for assessing study quality may have introduced selection or interpretation bias. The choice and inclusion of sources, based on thematic relevance rather than a structured quantitative analysis and strategy, limits the generalizability of the collected data. A second important limitation, also related to the study type, concerns the heterogeneity of research designs among the included studies. Evidence comes from heterogeneous sources differing in methodology, outcomes, sample size, and setting. This heterogeneity reduces the possibility of drawing robust and comparable conclusions, mainly for the substantial heterogeneity in dietary protocols, use of exogenous ketone supplementation, educational support, and participants’ clinical characteristics further limits the comparability of results. Taken together, these aspects suggest that the current evidence should be regarded as preliminary, and underscore the need for larger, rigorously designed randomized controlled trials with longer follow-up to confirm and extend the observations reported here.

## 5. Conclusions

This integrative review examined the impact of the ketogenic diet on the quality of life of patients with PD, with particular attention to non-motor symptoms such as depression, anxiety, and cognitive impairment, which are key determinants of health-related QoL. While some preliminary evidence suggests potential benefits, especially in the psychological and cognitive domains, the findings remain inconsistent and limited by methodological weaknesses, small sample sizes, and short follow-up. The KD appears feasible in the short term, with generally mild and transient adverse effects, but its long-term safety and effectiveness remain uncertain. At present, the KD should be considered an experimental and non-standard approach in the management of PD. Future large-scale, rigorously designed RCTs with standardized dietary protocols and longer observation periods are needed to clarify its role and determine whether it can provide a meaningful and sustainable improvement in QoL for these patients.

## Figures and Tables

**Figure 1 nutrients-17-03343-f001:**
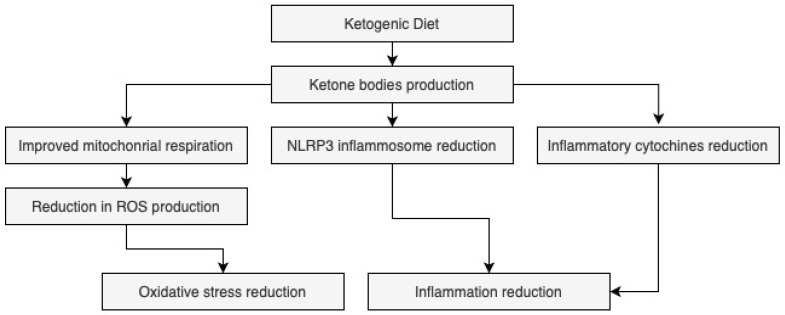
Effects of Ketogenic Diet on neuroinflammation and oxidative stress. *Legend*. ROS: reactive oxygen species; NLRP3: NOD-like receptor (NLR) family, pyrin domain containing 3.

**Figure 2 nutrients-17-03343-f002:**
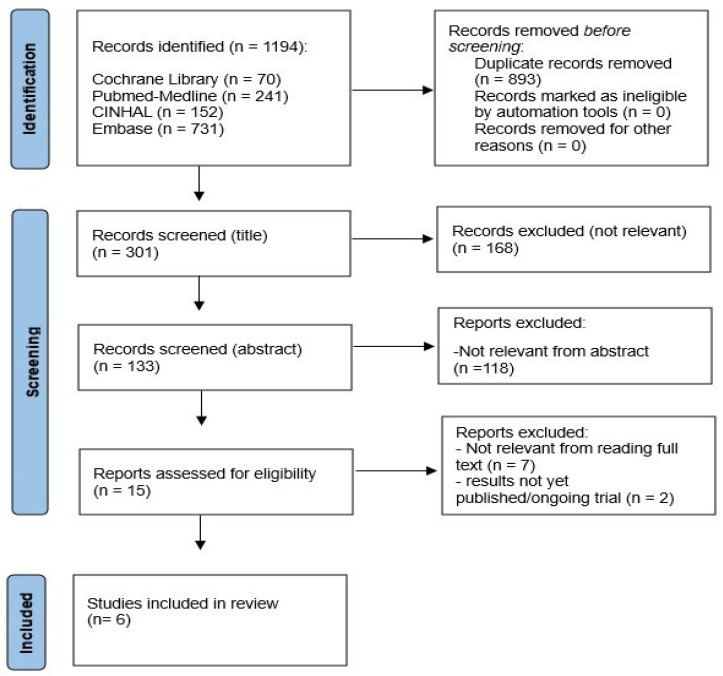
Prisma Flow Chart. *Legend*. updated to reflect the number of records from each database and reasons for exclusion.

**Table 1 nutrients-17-03343-t001:** Characteristics of the included studies.

Study	Design	Population	Intervention and Duration	Primary Outcomes	Outcome Measures	Study Limitations	Quality Rating
Choi et al. (2024), USA [33]	RCT (2-phase)	n = 16 Phase 1: IG = 7 CG = 9 Phase 2: IG = 16	Phase 1: Hospital-based (6 days) IG: KD + MCT CG: Standard diet Phase 2: Home-based (2 weeks) IG: KD + MCT total duration: 3 weeks	KD feasibility (retention, adherence, acceptability); Motor function effects; Metabolic biomarker changes; Cognitive and mood assessment	TUG; MDS-UDRS; NMSS; Metabolic biomarkers (BHB, insulin, triglycerides, HDL, HbA1c); EEG; DOPAC	Small sample size; Short study Duration; Lack of long-term follow-up; Potential placebo effect	++/Low
Tidman et al. (2024), USA [34]	Longitudinal study	n = 7	Intervention: KD Duration: 6 months	Motor and non-motor symptom improvement; QoL enhancement; Anxiety and depression reduction; Metabolic biomarker improvement	MDS-UPDRS I-II; CESD-R-20; PAS; Metabolic biomarkers (BMI, HbA1c, triglycerides, HDL, fasting insulin); H-Y Scale	Small sample size; Absence of control group; Possible selection bias and self-report bias; COVID-19 pandemic effects on measurements	++/Medium
Tidman et al. (2022), USA [35]	Quasi- experimental study	n = 16	Intervention: LCHF/KD Duration: 12 weeks	Health biomarker improvements; Anxiety symptom reduction	CESD-R-20; PAS; MDS-UPDRS; Blood tests (HbA1c, triglycerides, HDL, insulin, CRP); Anthropometric measurements	No control group; Small sample size; Possible placebo effect; Self-assessment of symptoms	+++/Medium
Koyuncu et al. (2021), Turkey [32]	RCT	n = 74 IG = 37 CG = 37	IG: KD CG: Standard diet Duration: 3 months	Voice quality improvements	VHI-10	Small sample size; Need for further research on pathophysiological mechanisms	++/Medium
Krikorian et al. (2019), USA [31]	RCT	n = 14 IG = 7 CG = 7	IG: KD CG: High carbohydrate diet Duration: 8 weeks	Cognitive performance improvement (lexical access, memory); Metabolic parameter changes	COWA; CVLT; VPLT; MDS-UPDRS; Metabolic analyses (glucose, insulin, D-BHB)	Small sample size; Limited duration; Inability to assess CNS biomarkers; Gender imbalance (predominantly male)	+++/Low
Phillips et al. (2018), [30] New Zealand	RCT	n = 44 IG = 22 CG = 22	IG: KD CG: Low-fat diet Duration: 8 weeks	Motor and non-motor symptom improvement; Metabolic parameter changes	MDS-UPDRS (I–IV); Metabolic parameters (weight, BMI, lipids, HbA1c)	Small sample size; Limited duration; Possible placebo effects; Adverse effects (worsened tremor and stiffness in some patients)	+++/Low

*Legend*: KD = ketogenic diet; LFD = low fat diet (high carbohydrate); LCHF = low carb high fat diet; HCD = high carb diet; SD = standard diet; D-BHB = β-Hydroxybutyrate; MCT = Medium chain triglycerides; RCT = randomized controlled trial; IG = intervention group; CG = control group; QoL = quality of life; MDS-UPDRS = Movement Disorders Society-Unified Parkinson’s Disease Rating Scale; COWA = Controlled Oral Word Association; CVLT = California Verbal Learning Test; VPLT = Verbal Paired Associate Learning Test; CESD-R-20 = Center for Epidemiologic Studies Depression Scale Revised; PAS = Parkinson Anxiety Scale; VHI-10 = Voice Handicap Index UPDRS = United Parkinson’s Disease Rating Scale; TUG = Timed Up & Go; NMSS = Non-Motor Symptom Scale; KPS = King’s Pain Scale; PDQ39 = Parkinson’s Disease Questionnaire-39; PSS = Parkinson’s Sleep Scale; MoCA = Montreal Cognitive Assessment; rsEEG = Electroencephalography; H-Y = Hoehn and Yahr Scale; Quality rating by JBI score [27,28,29]: +++, >70% (high quality); ++, between 70% and 50% (medium quality).

**Table 2 nutrients-17-03343-t002:** Ketogenic Diet and Assessed Parameters.

Study	Non-Motor Symptoms (MDS-UPDRS/UPDRS Part I)	Motor Symptoms (MDS-UPDRS/UPDRS Parts II, III, IV)	Body Weight	Lipid Profile	Cognitive Function	Quality of Life	Safety and Tolerability	Additional Parameters
Choi et al. (2024) [33]	Improved ↑***	No change =	Decreased ↓†	Triglycerides ↓† HDL ↑†	3-back test (working memory) =	Improved ↑***	Side effects: mild	TUG =; NMSS ↑; Continuous blood glucose ↓; HbA1c ↓†; Plasma dopamine =; DOPAC =; BHB ↑; CRP =
Tidman et al. (2024) [34]	Improved ↑	No change =	Decreased ↓	Triglycerides ↓ HDL =	PAS ↑ CESD-R-20 ↑	Improved ↑	Side effects: mild	BMI ↓; Waist circumference ↓ Blood glucose = HbA1c ↓ Insulin ↓* CRP = H-Y Scale ↓
Tidman et al. (2022) [35]	Improved ↑	No change =	Decreased ↓	Triglycerides ↓* HDL =	No significant cognitive improvement, but psychological improvement: PAS ↑ CESD-R-20 =	Improved ↑	Side effects: mild	Waist circumference ↓ HbA1c ↓ BMI ↓ Fasting insulin ↓ CRP =
Koyuncu et al. (2021) [32]	NA	NA	NA	NA	NA	Improved ↑	Side effects: mild	VHI-10 ↑
Krikorian et al. (2019) [31]	NA	No change =	Decreased ↓	NA	COWA (Lexical Access) ↑ CVLT and VPAL (Verbal Memory) ↑ CVLT (Mnemonic interference) Tendency to reduce	No change =	Side effects: mild	Waist circumference ↓* Fasting blood glucose = Fasting insulin = D-BHB ↑ Finger tapping =
Phillips et al. (2018) [30]	Improved ↑	No change =	Decreased ↓†	Triglycerides = HDL ↑ LDL ↑ Total cholesterol ↑	Improvement in fatigue and daytime sleepiness	Improved ↑	Side effects: mild	BMI ↓† Fasting glycemia ↓ HbA1c ↓† Urate ↑ BHB ↑ CRP =

*Legend*: “↑” = improvement (*p* value < 0.05) “↓” = worsening (*p* value < 0.05), “=” = neutral effect, “†” = no significant difference with the control group, “NA” = not analyzed. “*”= tendency but no statistical significance (*p* value > 0.05), “***” = very high level of significance (*p* value < 0.01), HDL: High Dose Lipoprotein; LDL: Low Dose Lipoprotein; UPDRS: United Parkinson’s Disease Rating Scale; MDS-UPDRS = Movement Disorders Society-Unified Parkinson’s Disease Rating Scale; PAS = Parkinson Anxiety Scale; CESD-R-20 = Center for Epidemiologic Studies Depression Scale Revised. **Note:** Values reported refer only to data collected in subjects undergoing KD. Quality of life was determined either qualitatively based on feedback provided by participants or as result of UPDRS part-1.

## Data Availability

Data supporting this research are available in this manuscript and Appendix A.

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
