# Peer review of "Effects of Ketogenic Diet on Quality of Life in Parkinson Disease: An Integrative Review"

_nutrients, 2025, doi:10.3390/nu17213343_

Round 1
Reviewer 1 Report
Comments and Suggestions for Authors
Manuscript: Effects of Ketogenic Diet on Quality of Life in Parkinson Disease: a Integrative Review
This integrative review synthesizes six studies (n≈152; 3–24 weeks to 6 months) on KD in PD. The paper’s key claims are: (i) small–moderate, but inconsistent, gains in QoL driven mainly by non-motor domains (depression, anxiety, fatigue, sleep; isolated cognitive signals), (ii) weak and heterogeneous effects on motor symptoms, potentially confounded by weight loss/placebo, (iii) consistent weight/BMI reduction, mixed lipids, and some glycaemic improvement, (iv) good short-term feasibility/adherence under structured support, with mild, transient AEs; long-term safety unknown. Overall, the topic is timely and clinically relevant, but conclusions are constrained by small samples, heterogeneity, short follow-up, and moderate–high risk of bias. Major revisions are required to improve transparency, analytic rigor, and clinical interpretability.
Major Strengths
- Addresses a clinically meaningful gap (non-motor symptom management in PD).
- Integrates clinical outcomes with metabolic and feasibility/safety data.
- Transparent acknowledgment of preliminary evidence and limitations.
- Uses JBI tools and an integrative review framework; protocol reportedly registered.
Core Limitations
- Small and heterogeneous evidence base: 6 studies; mixed designs (RCTs, pilot, longitudinal), varied KD protocols (classical, MCT-supplemented, LCHF, modified Atkins), comparators, and outcomes; meta-analysis not attempted.
- Risk of bias: open-label/pre–post designs, lack of blinding, short durations.
- Clinical ambiguity: motor benefits minimal; non-motor gains may be non-specific (weight loss/expectancy).
- Safety and implementation gaps: uncertain long-term renal/hepatic and cardiometabolic safety; real-world feasibility uncertain without intensive support; underexplored interactions with levodopa and other PD meds; risk of malnutrition/sarcopenia.
Required Revisions and Questions for the Authors
Based on a critical analysis of the provided integrative review, here are the main questions that could be addressed to the authors to improve the paper's clarity, rigor, and clinical interpretability. These questions are derived from the methodological details and limitations presented within the document itself.
1. Regarding Search, Selection, and Quality Appraisal:
Question: The search dates are listed as "November 2024 to January 2025" and the abstract states a search "up to June 2025". Could you clarify the exact end date of the literature search to ensure the review's currency?
Question:The paper mentions using JBI critical appraisal tools for RCTs, pilot studies, and longitudinal studies. Could you elaborate on why a quasi-experimental tool was used for some studies and how the final quality scores (e.g., "++/Medium") were derived from the JBI checklists, as this process is not fully transparent?
Question: The Prisma Flow Chart (Figure 2) shows the study selection process, but it omits the specific number of records identified from each database and the reasons for excluding full-text articles. Could this be detailed for greater transparency in the selection process?
2. Regarding Data Synthesis and Interpretation:
Question: Table 2 summarizes outcomes with symbols (↑, ↓, =), but often lacks specific quantitative data (e.g., mean differences, p-values for all mentioned outcomes). Could you include more specific quantitative results from the original studies to allow for a more nuanced interpretation of the magnitude of effects?
Question: The review concludes that non-motor symptom improvements are a key finding. However, you also state these gains may be due to "nonspecific factors such as weight loss or placebo responses". Did any of the included studies attempt to control for these confounders (e.g., through statistical analysis)? A more detailed discussion on this point would strengthen the clinical interpretation.
Question: The review notes inconsistent lipid responses, with one study showing an increase in LDL and total cholesterol. Given the cardiovascular implications, could you provide more detail on the quality of fats used in the different KD protocols (e.g., saturated vs. unsaturated) and discuss how this might explain the variable lipid outcomes?
3. Regarding Clinical Application and Safety:
Question: You highlight the risk of sarcopenia due to lean mass loss as a concern. Did any of the included studies specifically measure body composition or implement strategies (like adequate protein intake or resistance exercise) to mitigate this risk? This is a critical point for real-world application.
Question: The paper mentions that interactions between the Ketogenic Diet and PD pharmacotherapy are an "underexplored issue". Beyond the single study on levodopa pharmacokinetics, what specific theoretical concerns exist (e.g., related to protein competition for absorption, gut motility changes), and what should clinicians monitor for in practice?
Question: Feasibility was high in controlled settings with intensive support. Based on the reviewed studies, what were the most critical components of this support (e.g., caregiver education, cookbooks, MCT supplementation), and how might these be translated into a more scalable, real-world clinical or community setting?
Addressing these questions would significantly enhance the review's transparency, analytical depth, and practical relevance for clinicians and researchers.
Author Response
Dear Peer,
thank you for your time and effort. We hope you'll appreciate the manuscript revised according your relevant comments.
All Authors

Reviewer 2 Report
Comments and Suggestions for Authors
The paper presented by Golob et al. is a review aimed to impact of ketogenic diet on Parkinson’s disease. The authors conducted a detailed search of the literaturę and decided to use only valuable positions. In my opinion, the paper fits the Journal. Some small modyfications should be provided. Please see below my detail comments:
Introduction: provide mostimportant informaion; based on up to date references
Methods: authors decided to povide detail information about literature searching and quality of used study results. This point is well prepared.
Results: this section seems to be short and insufficient. Nevertheless, authors provided detail explanation why the number of presented sudies is so limited. In my opinion, it is acceptable. I would rework or change the layout of the tables 1 and 2 because it may be difficult to interpret. Perhaps simply changing the orientation to horizontal would be sufficient.
Discussion: sufficient, based on up to date references
Conclusions: provided; include most important information
Author Response

(The authors gave the same response as above.)

Reviewer 3 Report
Comments and Suggestions for Authors
The manuscript clearly states an aim: to examine the ketogenic diet (KD) in Parkinson’s disease (PD), specifically focusing on quality of life (QoL) and symptomatology. Also, the inclusion of both motor and non-motor symptoms, as well as secondary outcomes like tolerability and adherence, provides a comprehensive scope.
The definition of “quality of life” is vague. Is it measured via validated scales (e.g., PDQ-39, SF-36)? It would strengthen clarity to specify tools used.
The term "small to moderate effect size" is used without quantification—effect sizes (e.g., Cohen’s d) should be explicitly stated or at least exemplified.
A major limitation is lumping together studies with varying methodologies (e.g., different durations of KD, dietary compositions, and outcome measures). This makes synthesis less meaningful.
The small sample size (n=152 across 6 studies) raises questions about statistical power and the generalizability of conclusions.
The review fails to clearly differentiate between controlled trials and observational studies, which may differ significantly in quality.
Why do some studies report benefits while others do not? Are there differences in study design, KD formulation, adherence, or disease stage?
The severity and duration of adverse effects are not discussed. Were they self-limiting? Did they impact adherence?
There is no mention of metabolic lab monitoring, which is particularly relevant in KD (e.g., lipid profile, glucose, ketone levels).
The conclusion is appropriately cautious, but could be strengthened by offering specific recommendations for future research, such as:Standardizing KD protocols.Focusing on long-term effects.Stratifying patients by PD stage or phenotype.Incorporating biomarkers (e.g., inflammatory markers, neuroimaging).
Author Response

(The authors gave the same response as above.)

Round 2
Reviewer 1 Report
Comments and Suggestions for Authors
The paper can be accepted for publication in its current form, as it represents a well-conducted integrative review that transparently acknowledges both its own limitations and those of the included primary studies. The main opportunity for future improvement would be to move beyond a narrative synthesis toward a more quantitative and prescriptive analysis that could better guide future research and clinical practice.